# OpenReview forum: "Rule Based Rewards for Language Model Safety"
_NeurIPS.cc/2024/Conference — NeurIPS 2024 poster_

### Official Review · Reviewer_QvaG · 2024-06-13

**Soundness:** 3
**Presentation:** 2
**Contribution:** 4
**Rating:** 6
**Confidence:** 4

**Summary:**

This paper proposes an innovation to reward modeling, where rule-based rewards are added to a helpful-only reward model. These rule-based rewards have only a small number of parameters that can be fit with a small amount of data. This is meant to help with fast-changing behavior policies for LLMs, to update the LLM faster and with less pain. Additionally, it is meant to lead to more accurate rewards than could be achieved by rankings with human evaluators, especially if there are many rules to follow.

**Strengths:**

Overall, the paper seems highly original, has signs of high quality (lots of thought seems to have gone into their terminology, experimental designs, ablations, etc.), and significance (the work helps to automate part of the feedback to LLMs that are hard for humans to produce, thus making it easier to create nuanced and complex behavior policies for LLMs to follow).

**Weaknesses:**

The main weaknesses concern the clarity of the submission. I am basically sold on the paper, in the sense that it contained many soft signs that made me update toward "this is an important paper", thus leading to an accept score. However, it is only a *weak accept*, ultimately because I found parts of the submission quite hard to understand.

**1. Terminology:** The paper has a large amount of terminology with well-known words that have a very specific meaning in the context of this paper. I found it hard to keep it all together in my head. Examples of terminology are:
- content policy
- content areas
- content categories
- behavior policy
- completion type
- hard refusal, safe refusal, comply
- safety data
- rule-based rewards
- proposition
- feature
- rules

I appreciate that the authors highlight new terminology by using boldface. It might help to additionally create a diagram or other means to make the exact relationship of these apparent and to make it easy to keep it in ones working memory. Potentially the writing style could also help with this, by making the language less abstract or periodically using quick examples for reminders.

**2. Clarity in feature extraction:** It is a bit unclear to me whether the feature extraction only extracts features in completions, or also features to categorize the data into content categories. To explain my confusion: Ultimately, the features are proposition-truthiness values. So I looked up the propositions, and found the tables 5 and 6. These only contain propositions for completions, and nothing about the content categories! However, this confuses me: The rule-based reward is just a linear combination of features. But how are you supposed to compute a proper reward when the features do not specify the content category, meaning it is not clear (from the features alone) how the model should respond in the first place?

**3. Clarity on ranking of completions:** In Section 4.2, to write down the loss function, it is essential to know how the completions are ranked relative to each other. It is unclear to me how this is done. The authors write:

*first, use the content and behavior policy rules to determine rankings among completions based on their proposition values.*

So I assume what's happening is roughly the following: First, the feature extractor is used to determine the proposition values. If the content category is also known (possibly also through the feature extractors? This was my previous confusion, see above), then the behavior policy specifies a completion type, which comes, according to table 6, with a rule (i.e., composition of true/false values for features in the completion) that should be present. This is similar in type to Figure 2, but much more complex. Now, what's unclear to me: If you compare to completions, and they have the correct truth-values for some of the propositions but not for others, how exactly do you rank the completions relative to each other? One hypothesis I have is that you somewhere (but not in the paper, I think) have a mapping of combinations of proposition-truthiness values to the categories "ideal", "less_good", and "unaccaptable" that are in Figure 2, and then you only rank based on these three tiers (presumably with equality in the ranking if two completions are in the same tier? How is this handled in the loss then?), but I am not sure. An explanation would be helpful.

**Questions:**

**a.** I wonder whether this work could additionally help to alleviate the problems with distributional shifts in classical reward modeling: usually, you have the problem that when the LLM is optimized against the reward model, the LLM output-data is less and less like the data the reward model was trained on, which can lead to reward overoptimization.

I think this problem should be less severe in your work since I would guess the feature extractors to be more robust to distributional shifts than reward models. If the weights in the RBR are interpretable, then this should lead to more overall robustness.

What do the authors think about this point?

**b.** What's the connection between the content and behavior policies to [OpenAI's model spec](https://cdn.openai.com/spec/model-spec-2024-05-08.html)? They seem obviously strongly related.

**c.** The authors write "The design of good prompts for feature extraction is nontrivial, and requires some iteration."
Could you say more on how the prompts are designed, also for reproducibility?

**Limitations:**

Throughout the paper, the authors address and acknowledge limitations. Additionally, they have a dedicated limitations section.

---

> ### Author Rebuttal · Authors · 2024-08-07
>
> We are grateful to the reviewer for the constructive feedback and for pointing out areas where the clarity of our submission can be improved.
>
> **Terminology and Conceptual Clarity:** Thank you for pointing out that the large amount of terms can lead to confusion. As mentioned in the general response, we will be sure to include additional tables (such as Table 2 in the rebuttal pdf) in the main text and appendix to clarify, as well as improve the clarity of writing in the main text.
>
> **Feature Extraction and Content Categories:** We apologize for any confusion regarding feature extraction. In our methodology, content categories for prompts are pre-labeled using an automated labeling method and are not determined via feature extraction. This auto-label is necessary for the RBRs as it directly defines the desired response type for a given PPO prompt. Therefore, the propositions can directly look at just the completion as the desired response type is known. This information is also necessary for human data collection - while the annotators label the content category, and use their content category labels to decide which behaviors are ideal and non-ideal - we still need the auto labeled content categories to curate the set of prompts sent to the annotators to ensure there is a good balance of prompts as a random sample of prompts will mostly contain “safe” comply prompts and only contain a very small percentage of unsafe prompts. We will make sure to clarify these details in a future version.
>
> **Ranking of Completions:** We also apologize for any confusion regarding ranking of completions and will be sure to clarify in a future version. To synthetically generate our weight fitting set, we start with the train split of our safety prompts and criteria for a set of  desired completions for each prompt. For example, for a prompt that should be hard refused, we can decide we want the following set of 4 completions: one perfect hard refusal (ideal), two bad completions with randomly sampled bad refusal traits, such as judgement and/or illogical continuation, and one completion that contains the requested disallowed content. Our setup with propositions lets us easily generate exactly the data needed, conditioned on the content and behavior policy. We can use the natural language descriptions we already have to prompt for diverse completions. The goal is to have synthetic completions representing an ideal completion, a few sub-optimal completions, and an unacceptable completion for every prompt.  For each desired completion, we iteratively synthetically sample a candidate completion from a prompted Helpful-Only model, and use our RBRs, the moderation model and other quality LLM filters to confirm it contains the desired traits (ex. we did indeed generate a judgy bad refusal) and resample if necessary. While these completions are not guaranteed to be perfectly what we want (ex. A bad refusal labeled as judgy may not actually be judgy), we find they are generally pretty accurate through a quick spot check of 20 completions per response type, response label combo. This way, we can use the fact that a completion we generated by prompting for ideal should be ranked higher than a completion we generated by prompting to be bad. We release some example weight fitting data that was generated using this procedure in `data/weight_fitting_data` and provide example code for using it in `weight_fitting_example.ipynb`.
>
> **Alleviating Distributional Shift Issues in Reward Modeling:** We agree that rule-based rewards may provide more robustness to many potential issues, such as distributional shifts! For example, we have found that RBRs are be more robust to issues with reward over-optimization which we plan to talk about in a future version (see Section **(5) Human Data Reward Over-Optimization** in the rebuttal pdf).
>
> **Connection to model spec:** We agree that it seems that OpenAI’s model spec could be an interesting and more complex situation with expanded scope to apply RBRs.
>
> **Prompt Design for Feature Extraction:**  To tune our prompts, we first label a “gold” set by hand that contains (prompt, completion) pairs and whether a each proposition is true or not. We use this to tune classification-prompts for auto-LLM proposition grading by hand until high accuracy is achieved on the set. We are happy to say that in our supplemental code release, we were able to provide the exact classification-prompts used in our experiments (`config/proposition_prompts.yaml`) as well as this gold set (`data/rbr_gold_data` for data and `analyze_RBR_gold_data.ipynb` for basic stats).

---

> > ### Comment · Reviewer_QvaG · 2024-08-08
> >
> > Thank you for your answers! These are all helpful.
> >
> > I will not change the score since this would require seeing a substantially updated version of the paper, but I increase my confidence from 3 to 4. I am hopeful that the authors can improve the presentation substantially in an updated version.

---

> > > ### Author Response · Authors · 2024-08-13
> > >
> > > Thank you so much for your insightful and helpful feedback! We really appreciate all your suggestions on places where the writing can be clarified. Even before the rebuttal period, we have been committed to improving the paper past the initial submission (e.g. getting the code release, running human evaluations, additional ablation experiment), and we definitely will continue this commitment by incorporating all the valuable suggestions in a future version.

---

### Official Review · Reviewer_JMNb · 2024-07-12

**Soundness:** 3
**Presentation:** 3
**Contribution:** 2
**Rating:** 5
**Confidence:** 3

**Summary:**

The paper presents a novel approach to enhancing the safety behavior of Large Language Models (LLMs) through Rule-Based Rewards (RBRs). The authors argue that fine-tuning LLMs with human preferences can improve their capabilities and safety but may lead to unintended behaviors due to underspecified instructions. To address this, they propose a system where rules defining desired or undesired behaviors are scored by an LLM grader and used as additional reward signals during reinforcement learning training. The paper demonstrates that this method is effective in achieving safety performance comparable to human feedback while reducing over-refusals. The authors also discuss the impact of different design considerations, such as grader model size and the integration of RBRs with the reward model.

**Strengths:**

Strengths: The paper introduces a unique approach by combining rule-based systems with reinforcement learning for enhancing LLM safety. The submission is technically sound, with claims supported by empirical results. The paper is well-organized and clearly written. The findings are likely to be valuable to both researchers and practitioners.

**Weaknesses:**

While the integration of rule-based systems with reinforcement learning is novel, the paper could further emphasize how this approach can be generalized to other types of large language models beyond those tested. Additionally, future work should address the alignment issues associated with Chain-of-Thought (CoT) prompting (e.g. problems and methods mentioned in [1-3]), exploring methods to ensure that CoT frameworks do not inadvertently propagate or amplify biases and errors.
[1] Turpin, M., Michael, J., Perez, E., & Bowman, S. (2024). Language models don't always say what they think: unfaithful explanations in chain-of-thought prompting. Advances in Neural Information Processing Systems, 36.
[2] Feng, K., Li, C., Zhang, X., Zhou, J., Yuan, Y., & Wang, G. (2024). Keypoint-based Progressive Chain-of-Thought Distillation for LLMs. arXiv preprint arXiv:2405.16064.
[3] Yona, G., Aharoni, R., & Geva, M. (2024). Can Large Language Models Faithfully Express Their Intrinsic Uncertainty in Words?. arXiv preprint arXiv:2405.16908.

**Questions:**

1.	How do the authors ensure that the rules used in RBRs cover a comprehensive range of safety concerns without becoming overly complex or cumbersome to manage?
2.	How does the model handle conflicting rules?

**Limitations:**

Yes

---

> ### Author Rebuttal · Authors · 2024-08-07
>
> We thank the reviewer for their insightful comments and suggestions.
>
> **Generality of the Approach:** Our integration of rule-based systems with reinforcement learning is designed to be model-agnostic, and we believe it should generalize to other LLMs. In a revised version, we will include an additional plot (Figure 1 in the rebuttal PDF) that illustrates the RBR works on a variety of RMs with different tendencies. Including additional experiments to show it also can correct an RM trained with old, outdated data.
>
> **Chain-of-Thought (CoT) Prompting and Alignment Issues:** In our current work, we actually do not employ CoT to design the auto-LLM classification prompts for propositions. We did test using CoT for this, however we found it did not improve performance when using a ‘Large’ sized grader, most likely because our propositions were already broken down into simple binary tasks which were easy enough to classify. Additionally, for all propositions, we prompt tune and evaluate using a human labeled gold set (released with our code in the global rebuttal: see `data/rbr_gold_data` for the dataset and `analyze_RBR_gold_data.ipynb `for basic stats of the dataset) to further mitigate auto-LLM errors. We predict that CoT will be an important strategy when we move to other domains that have more complexity so we thank the reviewer for raising this important direction of future work to keep in mind.
>
> If the comment is about how RBRs can be used to improve the properties of CoTs, this is an interesting question. Our approach is agnostic to the prompts and response, and could be applied to any property that a grader LLM can evaluate effectively.  It would be interesting future work to explore applying such an approach to questions like faithfulness of CoTs.
>
> For evaluations, we test both with and without CoT and report average scores, we are open to providing more granular evaluation results separating CoT and non-CoT results if it would reduce concerns.
>
> **Comprehensive Rules in RBRs:** To cover a comprehensive range of safety concerns without becoming overly complex, we adopt a modular approach to rule design. Each rule addresses a specific aspect of safety policy, and the framework allows for the incremental addition or modification of rules as needed. This modularity ensures that the system remains manageable and adaptable to evolving safety requirements.
>
> **Handling Conflicting Rules:** Our framework ideally works with rules that are mutually consistent. Currently, any conflicting rules are handled at the behavior policy design level. If we accidentally design conflicting rules, we will have to resolve at the policy level and rerun the entire process - however because our approach relies on synthetic data, it is much faster than having to recollect human data. We will add discussion about this point in a future version.

---

### Official Review · Reviewer_CTB5 · 2024-07-12

**Soundness:** 3
**Presentation:** 2
**Contribution:** 3
**Rating:** 6
**Confidence:** 3

**Summary:**

The paper proposes a method for quickly incorporating safety mechanisms (i.e., refusal to obey the query) in LLMs. Given that production-use cases of LLMs can result in rapidly evolving safety requirements, the paper proposes a mechanism to incorporate a safety-based reward function into the LLM's RLHF / RLAIF step. The key idea includes defining the safety requirements via a set of binary propositions and logical compositions of these, called a rule, for each type of desired LLM response (e.g., one rule for situations where the LLM should refuse to comply, another rule for where it should refuse to comply with an apology, etc.). Given these rules, a set of LLM-based binary feature extractors can be defined as a text classification task. After sourcing completion preference (ranking) data for the updated safety requirements, either real or synthetic, a simple linear model can be fitted, reducing data needs. This now serves as a safety-based reward model. Incorporating the safety-based reward alongside the regular RLHF reward during LLM training allows the LLM to satisfy safe user requests at a higher rate, while properly handling unsafe user requests. This is demonstrated via experiments on safety datasets (XSTest, WildChat) as well as other capability (task) datasets (MMLU, GPQA, etc.).

**Strengths:**

+ The paper proposes an elegant, human-friendly mechanism to quickly incorporate evolving safety requirements into LLMs. This is now a large application area and safety remains a serious issue. Smaller organizations are often under serious time and budget pressures so anything that makes it easier to incorporate safety requirements is likely to have a large impact.

+ The primary contributions of the paper are algorithmic and empirical. The proposed algorithm for incorporating safety via an additional reward model optimized for a ranking objective is elegant and intuitively clear. The overall approach seems to have sufficient novelty (although a bit more clarity here would be nice).

+ The results indicate that the method seems to work well at improving safety without losing performance or refusing valid requests, while reducing data needs.

**Weaknesses:**

- The proposition-based RBR (safety reward model) has some novelty but also a lot of overlap with Sparrow. The other aspects of LLM training (SFT, RLHF, PPO, etc.) are also not novel. A more detailed description of the novel contributions of this work compared to prior work, both algorithmic and experimental, would help the reader place this work in the larger, fast-moving prior work involving LLM training and safety.

- The experimental section leaves a few important questions unaddressed. These include the error rate of the grader LLMs when scoring propositions, the data quality of the synthetic preference data and root causes for over-refusals. Given that this is related to safety, a careful error analysis would strengthen the paper. Important implementation details (Grader LLM, computational details) to aid reproducibility seem to be missing.

- Starting in Section 6, the description starts to get a bit cluttered in places. There's a lot going on in terms of datasets (Human, Synthetic, Helpful, Safety, etc.) and the differences in training and data between the resulting experimental variants (e.g., Table 3 in the appendix) got a bit confusing at times. A table summarizing each variant's key details (datasets, model training) would have made the paper easier to read.

**Questions:**

- How often do the grader LLMs make serious errors? Please characterize the grader LLM quality (feature noise) and implementations (e.g., prompts, if any) in more detail.

- What's the data quality of the synthetic preference data? How does it compare with the human preference data?

- What's the impact of using the restricted linear RM vs a bigger, non-linear RM? Does it make sense to consider a larger RM given the lower cost of synthetic preference data?

- What explains the higher over-refusal rate involving Human-PPO and Human-matchRBR-PPO?

**Limitations:**

Yes

---

> ### Author Rebuttal · Authors · 2024-08-07
>
> We thank the reviewer for the thorough evaluation and constructive feedback.
>
> **Relation to Prior Work and Novel Contributions:** Our work builds upon and extends concepts from prior studies such as Sparrow, particularly in using rule-based approaches for enhancing safety in LLMs. However, our approach introduces several novel contributions:
> - Sparrow focused on using human feedback - we instead focus on utilizing AI feedback. Sparrow was trained on 14.5k conversations generated and labeled by humans, whereas our approach required only human labeling for ~500 synthetically generated conversations. We hope that this cost, effort, and time reduction is impactful for the field.
> - To integrate their Rule RM with their preference RM, they utilize a fixed combination method. In contrast, our approach involves fitting a model to ensure that the final reward effectively and correctly ranks completions.
> - Lastly, we skip the step of distilling rules into RM data, a process that we found could be finicky ( see section **(5) Human Data Reward Over-Optimization** in the rebuttal PDF) and focus on incorporating the rules as directly as possible into PPO training.
> We will further elaborate on the differences between Sparrow and other related work in a future version of the paper.
>
> **AutoLLM grading accuracy:** We provide the accuracy per proposition of the LLM grader in `Table 7` in the appendix of the original submission. In a future version, we will add a compressed version of this table averaged across all propositions to the main text and make more references to the full table in the main text. Additionally, for more reproducibility, in our code release (see global rebuttal), we release the gold dataset used to score the proposition accuracies in `data/rbr_gold_data` (see` analyze_RBR_gold_data.ipynb` for basic stats of the dataset).
>
> **Grader LLM prompts:** We are happy to be able to provide the exact classification-prompts for the propositions we used in our code and data release in `config/proposition_prompts.yaml`. We used a “large” sized helpful-only SFT engine as a grader engine.
>
> **Synthetic Data Quality:** The labels of the synthetic data are most likely noisier than the human labeled gold data set as they are synthetic and have not gone through human verification. We spot-checked 20 examples from each response-type (ex. hard-refuse) and response (ex. Perfect refusal, bad refusal, etc) and found the response labels were mostly accurate. We will add this analysis to the appendix in a future version. We additionally release example synthetic data generated using the procedure detailed with our code release (`data/weight_fitting_data`, see `weight_fitting_example.ipynb` for description and usage) to allow external parties to study its accuracy as well.
>
> **Other models for combining the RBR and RM:** We combine the scores given by our “large” sized helpful-only RM with the RBR scores by fitting a linear model of features. Note that “large” is the largest model size considered in these experiments. During the research process, we did plan on moving to non-linear models if necessary, however we found the performance of a linear model was already quite good based on the analysis given in `Figure 3` of the original submission. In the future if we extend this work to even more complex domains, we do anticipate possibly needing a more complex model (such as possibly fitting a small neural network).
>
> **Higher over-refusal Rate:** We believe the high over-refusal rate of human data could be due to a few possible reasons: We find annotators tend to be a bit more over-cautious in terms of assigning label categories (ex. For borderline refusal prompts, they would lean towards marking the prompt as hard_refuse). Additionally for human data, there is the question of balancing the RM data, (we include some discussion of this in the rebuttal PDF section **(5) Human Data Reward Over-Optimization**), we choose a 1-1 balance of hard-refusal and comply by default for human RM data, but it is possible that other ratios would result in different performance. One advantage of the RBR is that it removes the need to tune RM data balance, which we found to be a difficult parameter to tune.
>
> **Confusing Terms:** As mentioned in the global response, we will be sure to include your helpful feedback and include clarifying tables as well as clarifying the text in a future version.

---

> > ### Comment · Reviewer_CTB5 · 2024-08-13
> > **Re. author response**
> >
> > I thank the authors for their detailed response. After reading the response, other reviews and comments, I've increased my score to "Weak Accept". The paper could be much clearer than it is currently and the authors do commit to fixing this. However, as Reviewer QvaG noted, this is a rather large delta from the current version. This prevents me from increasing my score further.

---

> > > ### Author Response · Authors · 2024-08-13
> > >
> > > Thank you so much for your valuable feedback and suggestions, they are really helpful for improving the clarity of our work! We have been working to improve the paper past the initial submission, even before the rebuttal period began (e.g. getting the code release, running human evaluations), and will definitely continue to improve it by incorporating all your feedback in a future version.

---

### Author Rebuttal · Authors · 2024-08-07

We thank the reviewers for their thoughtful and constructive feedback. In this global response, we aim to inform reviewers of some additional content we provide and respond to common themes. We include this additional information in the rebuttal PDF.

**(1) Code and Data release:** We release some code and data at the following anonymized link: https://anonymous.4open.science/r/Safety-RBR-Code-Data-Anonymous-0183/readme.md

We provide:
- Our final classification-prompt for each proposition we used (`config/proposition_prompts.yaml`).
- Code for fitting the linear model for combining RBRs with the RM and generating the reward distribution histograms and error rates (`weight_fitting_example.ipynb`).
- Example synthetic data for weight fitting and evaluation generated using the synthetic data procedure described in the text (`data/weight_fitting_data`).
- Example RM data (`data/weight_fitting_data/rewards`).
- The RBR gold set we used for evaluating the auto-grading procedure and tuning prompts for auto-grading (`data/rbr_gold_data`) along with a notebook that calculates basic stats (`analyze_RBR_gold_data.ipynb`).

**(2) Human Evaluation:** We also have completed human evaluations and we provide this updated information in `Table 1` of the rebuttal pdf

**(3) Terminology and Clarity:** Our first and third reviewers provided helpful feedback that the various terminology and datasets may have contributed to some confusion. We will add a summary table of datasets (`Table 2` in the rebuttal pdf) to the main text. We will also add similar additional tables to the appendix, including a comprehensive table of terms and their definitions and a table mapping propositions to class (ex. ideal) as suggested by the third reviewer, as well as a table summarizing each experimental variant's key details (datasets, model training) as suggested by the first reviewer.

**(4) Updated Results Figure:** We provide a second results figure (`Figure 1`) in the results PDF to show that RBRs can generalize to RMs of different tendencies. We include a new experiment on an RM we trained that included old outdated data and show that it is helpful there as well.

**(5) Human Data Reward Over-Optimization:** We provide additional details on an interesting situation with reward over-optimization that occurred with the human data we collected, and how we corrected for it, along with discussion of how RBRs avoid this specific issue.

---

### Decision · Program_Chairs · 2024-09-25

**Decision:**

Accept (poster)

**Comment:**

This paper introduces Rule-Based Rewards (RBRs) for preference modeling of LLMs, which allows for precise behavior control and rapid updates using minimal human data. RBRs consist of rules for desired or undesired behaviors and are scored by an LLM grader to enhance reinforcement learning training.

All reviews are positive and acknowledge the novelty of the proposed method. The empirical study is also solid. However, as pointed out by reviewers, the presentation can be improved to enhance clarity. The authors should include the promised modifications in the final version.